# Shallow and deep learning of extreme rainfall events from convective atmospheres

Gerd Bürger[1], Maik Heistermann[1]

[1]Institute of Earth and Environmental Science, University of Potsdam, Potsdam, 14476, Germany

*Correspondence to:* Gerd Bürger (gbuerger@uni-potsdam.de)

**Abstract.** Our subject is a new Catalogue of radar-based heavy Rainfall Events (CatRaRE) over Germany, and how it relates to the concurrent atmospheric circulation. We classify daily ERA5 fields of convective indices according to CatRaRE, using an array of 13 statistical methods, consisting of 4 conventional ('shallow') and 9 more recent deep machine learning (DL) algorithms; the classifiers are then applied to corresponding fields of simulated present and future atmospheres from the CORDEX project. The inherent uncertainty of the DL results from the stochastic nature of their optimization is addressed by employing an ensemble approach using 20 runs for each network. The shallow Random Forest method performs best with an Equitable Threat Score (ETS) around 0.52, followed by the DL networks ALL-CNN and ResNet with an ETS near 0.48. Their success can be understood as a result of conceptual simplicity and parametric parsimony, which obviously best fits the relatively simple classification task. It is found that on summer days, CatRaRE-convective atmospheres over Germany occur with a probability of about 0.5. This probability is projected to increase, regardless of method, both in ERA5-reanalyzed and CORDEX-simulated atmospheres: for the historical period we find a centennial increase of about 0.2 and for the future period of slightly below 0.1.

## 1 Introduction

Pluvial floods and flash floods are among the most destructive and ubiquitous natural hazards in Central Europe (European Environment Agency, 2019). Both are caused by convective heavy rainfall: the first by local accumulation of surface runoff mainly in urban areas, the second by fast runoff concentration in head water catchments. While, in general, global warming is expected to increase the frequency and intensity of such events (Seneviratne et al., 2021, Table 11.17), the intricate nature of atmospheric convection makes it difficult to quantify trends. On the one hand, sufficiently long rain gauge records are rare, and the representative observation is difficult due to the spatial heterogeneity of convective rainfall. On the other hand, coupled global and regional climate models struggle with the physical and numerical complexity of the small-scale, short-term convective processes. While convection-permitting regional models are able to resolve such processes (Lucas-Picher et al., 2021; Fosser et al., 2020), their applicability at large spatial scales and long time

periods is still limited by computation cost. Resorting therefore to 'only' parameterizing convection for dynamical simulations is, for the time being, still loaded with uncertainty but without alternative (Lepore et al., 2021).

One possible way out of this dilemma is to establish an empirical relationship between the large-scale atmospheric state, as represented by global climate models, and the occurrence of convective heavy rainfall in any specific area of interest. Given a sufficiently robust relationship, this would allow us to use global reanalyses and future projections to infer possible trends in the past and future event frequency. This approach is part of a long and well-known history of bridging the gap between the coarse scales of global climate models and the small scales of many climate impacts (Doblas-Reyes et al., 2021, 10.3). From a machine learning perspective, though, the task itself can be framed as a straightforward classification problem: the classification of large scale atmospheric states with regard to their potential to produce extreme convective rainfall events.

It is in this specific context where our study aims to explore the ability of various machine learning architectures to solve this classification task. Since computing power has grown to levels that were beyond imagination just years ago, automated and numerically expensive (machine) learning has evolved into a versatile and capable tool set for data science. This applies in particular to Deep Learning (DL), which refers to neural networks with a notably increased number of neuron layers. Many scientists are now curious whether their older, conventional models can stand the test of skill against these newer methods. Examples are abundant, for example from climate simulations and weather prediction (daily to seasonal) (Gentine et al., 2018; Ham et al., 2021, 2019; O'Gorman and Dwyer, 2018; Rasp et al., 2018; Weyn et al., 2021; Schultz et al., 2021; Reichstein et al., 2019; Irrgang et al., 2021; Wang et al., 2022). Generally, DL is evolving with such a speed that makes it hard to keep pace; for a general introduction into Deep Learning, Bianco et al., (2018), Goodfellow et al., (2016), or Alzubaidi et al., (2021) provide overviews. At least in the data driven disciplines, hence, there is a chance that much of the scientific progress of the past several decades is about to be dwarfed by machine learning techniques. Our study explores the potential of DL using as a benchmark several state-of-the-art conventional methods, referred to here for lack of a better expression as *shallow* methods.

For this purpose, we use a recently published catalogue of extreme heavy rainfall events (CatRaRE, Lengfeld et al., 2021), contiguous in space and time, that were extracted from a 20-years record of gridded hourly radar-based precipitation estimates (RADKLIM, Winterrath et al., 2018). We use this list of convective events to "label" daily atmospheric fields from a reanalysis whose spatial resolution is coarse enough to permit long climate model projections. Given the limited sample size of two decades of daily labeled fields, and also to keep the possible ramification of results manageable, we decided to use only the single daily label which is "an event did or did not occur anywhere in Germany". If from the set of shallow and deep methods a useful classifier is found, the labeled atmospheric fields, from reanalyses and simulations, should provide present and future statistics about the occurrence of such CatRaRE-type events; for the decision maker, they represent a first stage of awareness around the future occurrence of those events over Germany.

By interpreting each atmospheric field as the color code of a 2-dimensional "image", our binary classification problem can be framed as one of image classification. Given the geometry and resolution of the fields (cf. section 2.1), the classification

is done in a space of dimension ~4k. This number roughly compares to some of the classical DL datasets such as MNIST (dim. ~1k) and CIFAR-10 (dim. ~3k), but is certainly small compared to newer sets such as ImageNet (dim. ~100k) or Open Images (dim. ~5M), cf. Table 2. Likewise, while most of the DL networks have to choose between as many as 1000 classes, our initial example is just binary. Therefore, if CatRaRE-relevant patterns of atmospheric moisture over Germany can be compared at all to images of cats and dogs, one could naively expect the classification performance to be at least as good as published results on those image datasets. And the prospect for using a more fine-grained analysis with more sub-regions (= more classes) should then, so we hope, be equally good.

Still, one should not overemphasize the narrative of cats and dogs: like atmospheric classification in general, our approach considerably simplifies the true relationship between the large-scale fields and the local-scale processes. Most notably, and quite unlike the animal examples, impacts exist on a entire spectrum of strength, and their occurrence is only registered as binary through the application of thresholds. Moreover, the spatial statistics which convolutional nets exploit so successfully (contour features such as edges and corners) do not in general transfer to atmospheric fields; nor do temporal statistics, because physical fields tend to be affected by instationarities such as trends.

After thorough evaluation of the results from reanalyses, the methods are applied to simulated atmospheres from the EURO-CORDEX project (Vautard et al., 2021); the predicted classification is used to estimate past and future changes in the frequency of extreme events as represented by CatRaRE. Our focus here shall generally not be on obtaining the best result currently possible, but rather on understanding the sensitivity to the various methods, and whether there is any merit from using the deep methods. To that effect, we explore a selection of DL architectures that had, each in its time, entered the DL arena quite spectacularly; an overview of the architectures is given in the Supplemental Information (SI). We attempt to understand if and why they perform differently for the case of CatRaRE over Germany.

The overall approach taken here resembles that of Ukkonen and Mäkelä (2019), who use a grid-based approach to classify thunderstorm activity from an array of up to 40 indices representing atmospheric instability, humidity, and inhibition. It deviates in a number of aspects: a) our target is a special database of impact-relevant convective rainfall cells; b) we avoid the imbalanced classification and correspondingly low skill of a gridpoint-based approach; c) we use much fewer predictors but deeper networks, and d) we apply the estimated classifiers to a future climate simulation.

To summarize, we explore here the performance of shallow and deep machine learning methods to classify large scale atmospheric fields with regard to their potential to produce CatRaRE-type convective rainfall events, and to use those classifiers to estimate past and future statistics of CatRaRE-type events.

## 2 Methods and Data

### 2.1 Atmospheric data

Since our focus is on convective events, we restrict the analysis to the warmer months from May to August. From the ERA5 reanalyses for the years 1979 to 2020 (Hersbach et al., 2020), atmospheric convectivity is measured by the indices of convective available potential energy (*cape*), convective rainfall (*cp*), and total column water (*tcw*). They are used as potential classifiers, given as daily averages over the area between the edges [5.75E 47.25N] and [15.25E 55.25N], normalized with, for each variable, mean and standard deviation across time and space. (For simplicity, non-normality of the indices and a more refined normalization via logit or probit was not taken into account). Future atmospheric fields are obtained from the EURO-CORDEX initiative and are simulated by the model CNRM-CM5 (simply "GCM" in this text) driving the regional model COSMO-crCLIM ("RCM") for the European area (EUR-11) (Sørland et al., 2021; Leutwyler et al., 2017); COSMO-crCLIM is run in 12 km horizontal resolution with parameterized convection and is the only model that provides *cape*; the grid is re-mapped to the ERA5 grid. We use emissions from both historic (1951–2005, "HIST") and RCP85 scenarios (2006–2100). The atmospheric fields are given as normalized anomalies, using mean and standard deviation of the reanalyzed and simulated fields from the common period 2001–2020 as a general reference state; for the latter, it requires to concatenate the corresponding sections from HIST (2001–2005) and RCP85 (2006-2020) to form the reference.

### 2.2 CatRaRE

We use the catalogue of radar-based heavy rainfall events  (CatRaRE, Lengfeld et al., 2021), which defines heavy rainfall based on the exceedance of thresholds related to warning level 3 of Germany's national meteorological service (Deutscher Wetterdienst; DWD hereafter); this corresponds to more than 25 mm in one hour or 35 mm in six hours (and roughly to a return period of 5 years, cf. SI). Based on threshold exceedance of individual radar pixels, heavy rainfall objects are constructed that are contiguous in space and time, and for which an extremeness index ($E_{T,A}$, *M*üller and Kaspar (2014)) is inferred that is a combined measure of area, duration and intensity. In this study, a day is labeled as *extreme* if the database contains an event for that day with $E_{T,A} > 0$ and of at most 9 hours duration; it means that somewhere in Germany a corresponding severe weather was recorded, and the limited duration serves as a rough proxy that the event was convective. The classification thus depends on the particular shape of the German border although its quasi-random details (an event might occur just outside Germany) have certainly no representation in the atmospheric predictor fields. This is the same kind of limitation that station downscaling shows in general, and cannot be avoided.

On average, 51% of the (May–Aug) days see such an extreme event, which means that, although CatRaRE events are locally rare by definition, the main classification task (event vs. no event in Germany) is quite balanced. Mainly for later use we counter any potential class imbalance nevertheless, and employ a rather simplistic oversampling approach by populating the minority class with random duplicates of that class until that class is no longer minor.

The ERA5 grid is shown in Figure 1, along with the average cape values for 28 July 2014. It was a day with particularly strong atmospheric convectivity that led to several severe rainfall events all over Germany. The events were monitored by CatRaRE, so that the day is labeled as extreme. Two active regions are visible, one in the Southwest and one in the central West. There, in the city of Münster, occurred the most disastrous event, with one station recording as much as 292 l/m² within 7 hours (Spekkers et al., 2017) The surrounding cape grids show values > 600 J/kg, similar to other areas in Germany (SE, NE).

## 2.3 Conventional ("Shallow") and Deep Learning models

**Table 1. The Shallow-Learning methods.**

|  | abbr. | note | source |
|---|---|---|---|
| Lasso logistic regression | LASSO | cross-validated penalty (14 predictors) | (McIlhagga, 2016) |
| random forests | TREE | 200 trees | (Jekabsons, 2016) |
| shallow neural nnet | NNET | 2 hidden layers with 7 and 3 neurons | Octave |
| logistic regression | NLS | nonlinear least squares | Octave |

As competitive benchmarks to DL models, we employ four shallow statistical models: Lasso logistic regression (LASSO), random forests (TREE), a simple neural net with 2 hidden layers (NNET), and logistic regression based on nonlinear least squares (NLS). All of these are applied with and without Empirical Orthogonal Function (EOF) truncation, using North's 'rule of thumb' to find 33, 27, and 21 principal component predictors for *cape*, *cp*, and *tcw*, respectively, as estimated from the calibration period; more details are listed in Table 1 and in the source code mentioned at the end.

The architectures of the selected DL models are almost exclusively based on *convolutional neural networks* (CNNs), a concept that was introduced with the famous LeNet-5 model of (LeCun et al., 1989) for the classification of handwritten zip codes. Besides LeNet-5 we use the network architectures AlexNet, ALL-CNN, CIFAR-10, GoogLeNet, DenseNet, and ResNet. These were created for the classification of digitized images, such as the CIFAR-10 set with 32×32 image resolution and 10 classes or ImageNet with 256×256 images covering 1000 classes, and regularly used in annual image classification contests since about 2010 (Krizhevsky et al., 2017). Along with these come two quite simplistic networks, *Simple* representing a single convolutional and a dense layer, and *Logreg* with just one single dense layer; they are used mainly for checking and benchmarking our *Caffe*-code implementation; details are provided by Table 2 and the SI. This provides a fairly comprehensive selection from the most simple to highly sophisticated networks. Details of the corresponding model implementations can be inspected directly in the code, see section 5.

**Table 2. The Deep-Learning architectures. The number of classes pertains to the reference study. We only count convolutional and fully connected (inner product) layers.**

|  | Year | resolution | layers | # parameters (·103) | Reference | Original classes |
|---|---|---|---|---|---|---|
| LeNet-5 | 1989 | 28×28 | 4 | 400 | (LeCun et al., 1989) | 10 |
| AlexNet | 2012 | 227×227 | 8 | 60000 | (Krizhevsky et al., 2017) | 1000 |

| | | | | | | |
|---|---|---|---|---|---|---|
| CIFAR-10 | 2014 | 32×32 | 4 | 80 | (Krizhevsky et al., 2017) | 10 |
| ALL-CNN | 2014 | 32×32 | 9 | 1000 | (Springenberg et al., 2014) | 10 |
| GoogLeNet | 2014 | 224×224 | 76 | 10000 | (Szegedy et al., 2015) | 1000 |
| ResNet | 2016 | 32×32 | 22 | 300 | (He et al., 2016) | 10 |
| DenseNet | 2016 | 32×32 | 159 | 1000 | (Huang et al., 2017) | 10 |
| Simple | | 32×32 | 3 | 300 | this paper | 2 |
| Logreg | | 32×32 | 1 | 6 | this paper | 2 |

Compared to the original DL classification tasks in the literature, with e. g. 1000 classes for AlexNet and GoogLeNet, cf. Table 2, our classification in its initial form is just binary, so naturally some of the network and solver parameters had to be adjusted. A crucial "hyperparameter" is the size of the training and testing batches *(batch_size* in Caffe*)*, which had to be lowered for the broader and deeper networks. Another parameter is maximum iteration (*max_iter*); unless that number is reduced drastically the optimization would enter a runaway overfitting process whose emergence is barely visible. In order to stabilize the stochastic optimization, the gradient search is increasingly damped based on a factor called the base learning rate (*base_lr*); the learning rate decay policy *poly*, which required a single parameter *power*, helped to steer the learning process in a parsimonious way; it was used for all DL solvers; the decay at iteration *iter* is governed by the formula *base_lr·(1 - iter/max_iter)^power*. All adjusted parameters are listed in Table S1 from the SI.

Because DL optimization generally uses a stochastic gradient descent algorithm and is therefore not fully deterministic, we use an ensemble of 20 DL optimization runs. This ensemble, too, is informative about network convergence, and in some cases even reveals potential for refined parameter tuning. All relevant details are described in the SI, section 2.

The predictor fields of *cape*, *cp*, and *tcw* are taken as three 'color channels' (RGB) of an image sequence. Because the image resolution differs between the networks, varying from 28×28 pixels for LeNet-5 to 227×227 pixels for AlexNet, a regridding of the fields is required to match the resolution of the original model, cf. Table 2. Except for LeNet-5, this represents an upsampling so that the pattern itself (its shape) enters the DL essentially unchanged (and the LeNet-5 resolution is sufficiently similar). EOF truncation was consequently not applied to the DL models.

Our machine learning framework is Caffe, which provides a genuine Octave/Matlab interface to DL (Jia et al., 2014). The Caffe framework along with most of the networks have already seen the height of their days, and are by now being superseded by more sophisticated and successful networks and frameworks (Alzubaidi et al., 2021). This only indicates that the development continues to be fast, making it difficult to keep pace.

### 2.4 Calibration, Validation

The full period from 2001 to 2020 amounts to a total of 2460 days, which we split into a calibration (train) and validation (test) period of 2001–2010 and 2011–2020, respectively. For the DL training, cross-entropy is used as a loss function. As evaluation measure the Equitable Threat Score (*ETS*, syn. Gilbert Skill Score) is used. ETS measures the rate of correctly forecast extremes relative to all forecasts except majority class hits, adjusted for random hits. We note that the validation

data are not completely independent of the DL models. Because they have been used for inspecting the learning curves and their convergence, there is a slight chance that the validation scores may reflect sampling properties and would therefore not generalize. On the other hand, the tuning goal was to achieve reasonable convergence of the loss function and not to minimize its value. Therefore, we are confident that overfitting is reasonably limited.

## 3 Results and discussion

### 3.1 Network training and testing

Convergence of the DL model optimization is exemplified in Figure 2, which depicts the crossentropy loss function during the training and testing (syn. calibration and validation) iterations. LeNet-5 follows a typical path of learning progress, with variable but decreasing loss for the training phase that is closely traced by the testing phase, the latter leveling out somewhat below a loss of 0.4. The learning curves of the other networks look similar but with different absolute losses, and are shown in Figure 3. It is noticeable that e. g. ResNet converges after only 40 iterations whereas AlexNet and ALL-CNN require, respectively, 500 and 1000 iterations. Also note that the simpler networks such as Simple, Logreg, and CIFAR-10 remain stable after reaching convergence while, what is not shown in the Figure, the more complex networks AlexNet, GoogLeNet and ALL-CNN do not and start to diverge, indicative of overfitting.

### 3.2 Classification performance

The probabilistic predictions are now transformed to binary (classification) predictions by choosing, from the calibration period, for each model the threshold that maximizes the *ETS*. Classification performance when driven by ERA5 fields from the validation period 2011–2020 is shown in Figure 4. Comparing the 3-channel predictors *cape*, *cp*, and *tcw* against the two channels *cp* and *tcw*, it confirms the influential role of cape as a predictor (Ukkonen and Mäkelä, 2019): it improves skill across all models, an exception being the poorly performing NLS model with no EOF truncation of the predictor fields; except for LASSO where it has little effect, EOF truncation generally improves shallow model skill. The scatter of DL model skill, crossentropy versus *ETS*, is indicative of the stochastic nature that is inherent in all DL results (Brownlee, 2018; see also Kratzert et al., 2019), and uncertainty obviously grows with network complexity. The best overall performance according to the Figure is achieved by the TREE method (*ETS = 0.52*), with several of the ALL-CNN and ResNet realizations coming close, nevertheless, so that on average these turn out second. The LeNet-5, Simple and CIFAR-10 networks reveal a stretched cloud with larger variation along the *ETS* axis. That this is not a simple scaling issue can be seen by comparing Logreg and LeNet-5, whose optimized crossentropy values show virtually no variation while *ETS* varies stronger. Crossentropy as a loss function, so it appears, sufficiently dictates unique convergence for the training phase, but apparently does not constrain the models enough to make good predictions for the testing phase. For logistic

regression (NLS), EOF truncation is indispensable as it otherwise leads to heavy overfitting. Stochasticity is not limited to DL, it is also contained in NNET as a 'normal' neural net and, as the name suggests, random forests (TREE); the scale of variation is much smaller, however. Like for the DL networks we form ensembles also for NNET and TREE, as further explained in the SI. And as Fig. S3 demonstrates, a second realization of the shallow and deep ensembles essentially yields similar results. In the following DL applications the ETS-optimal ensemble members are used.

Differences in DL model performance are difficult to interpret, but a few hints may be obtained by inspecting the network architecture. Quite roughly, the width of a convolutional network represents the number of learnable features whereas the depth measures the grade of abstraction that can be formed from these features. A convective atmospheric field is, compared to a landscape with cats or dogs in it, quite simple. If a network architecture scales well this simplicity should not matter. However, very rich architectures also require a wealth of data to learn their many parameters from (14M images in ImageNet), which we do not have here. Particularly the very wide and/or deep networks such as AlexNet, GoogLeNet, or DenseNet may suffer either from inferior scaling behavior or too little data. ALL-CNN and ResNet, on the other hand, are designed particularly for simplicity and parsimony (Springenberg et al., 2014; He et al., 2016), with good performance across a broad spectrum of applications and apparently best adapted to our case.

### 3.3 Probabilistic reliability and sharpness

We further illustrate the reliability of the probabilistic predictions by means of reliability diagrams, first in Fig. 5 for the shallow methods; these come with an inset forecast histogram, displaying the relative frequencies of the delivered probabilities as a measure of sharpness of the prediction. The methods are quite reliable, except that TREE's lower-probability predictions occur too rarely and the higher ones too often. LASSO and TREE predictions are, as the inset shows, moderately sharp, unlike NLS and especially NNET which is almost perfectly sharp. Most of the deep methods are reliable, cf. Fig. 6, exceptions being ResNet, DenseNet and GoogLeNet, whose predictions of medium probabilities occur too often. They are also more reliable than the shallow methods and generally sharper, especially CIFAR-10, GoogLeNet, and DenseNet with a high load of near yes/no predictions.

### 3.4 Model application

We now apply the trained models to the observed (reanalyzed) and simulated atmospheric fields. It means we obtain for each summer day from the corresponding atmospheric model period a prediction expressing the probability of a CatRaRE-type event happening somewhere over Germany. Starting with the ERA5 reanalyses, we check whether the July 2014 event is captured by the ERA5 fields. Figure 7 shows a typical probability forecast from the DL model LeNet-5. During the days in late July of 2014, there is permanent convective activity over Germany. LeNet-5 shows near-certainty predictions for events to occur, including the July 29 extreme event. Sporadic periods of little activity are also well reflected by LeNet-5.

For a broader temporal picture, we form annual (i. e. May–Aug) averages of the daily probabilities, and display the entire reanalysis period (1979–2020) in Figure 8. The classification is obtained from the best-scoring model TREE along with ALL-CNN. The observed CatRaRE climatology (2001–2020) shows a mean daily probability of 0.51, and it is well reproduced by both models. For the period 1979–2005, which is the common period for historic reanalyses and simulations, they reveal strong positive centennial trends of 0.34 and 0.35, respectively; the trends are significant using a

significance level of *α=0.05* throughout the study. (A linear trend is obviously only partly meaningful for a bounded quantity such as probability, but we use it here nevertheless.) Annual correlations are equally strong (0.65 for both); corresponding plots for all other models are altogether similar and are, for completeness, shown in Figs. S3 and S4; note, however, that the centennial trends are slightly weaker, as also shown in Table 3. Interestingly, the model with almost the poorest daily performance (*ETS = 0.44*), NLS, reveals the highest annual correlation of 0.62 with observations.

Now we analyze the CatRaRE classifications for the simulated atmospheres from past to future (1951–2100), based on HIST and RCP85. Again, we first turn to the overall best performing model TREE along with ALL-CNN, as shown in Figure 9. Both appear to be relatively unbiased (with respect to the normal period 2001–2020) and, as Table 3 shows, the HIST simulations exhibit positive trends (1979–2005) of around 0.16 and 0.15, respectively, which amounts to only half of what was seen for ERA5; for RCP85, only ALL-CNN exhibits a significantly positive centennial trend *of 0.07*. The trend

discrepancy between ERA5 and HIST is most pronounced for these two methods and weaker for the other, cf. Table 3, Figs. S5 and S6. The RCP85 trends are, except for TREE, significant but smaller, which may be the result of applying a linear trend to a bounded quantity. We checked whether the resulting trends differ significantly among the ensemble of 20 DL training and application runs; except for rare cases they do not, and they remain positive throughout.

Different annual correlations of the CatRaRE probabilities may contribute to the discrepancy between ERA5 and HIST

trends. But a direct inspection of the predictor indices *(cape, cp, tcw)* is indicated nevertheless, which we have done in Fig. 10. All predictors have positive trends, and all are significant except for present *cp*. While *cape* shows stronger interannual variations with some very high values later into the future, *tcw* displays a fairly steady increase that appears consistent over the entire time domain 1951–2100, reanalyzed and simulated; this is likely related to the increasing water-holding capacity of the warming atmosphere. The *cape*-trend is much stronger for ERA5, which may partly explain the trend

discrepancies of Table 3. Note, however, that upper-air measurements needed for *cape* from observations are sparse, and corresponding trends therefore uncertain (Taszarek et al., 2021).

In view of such discrepancies, a note of caution is finally in place. In our modeling approach we have tacitly assumed that the learned statistical relationships remain valid when applied to previously unknown atmospheres, and remain so even when those are from a dynamical simulation or a different climate. That this may indeed cause problems became apparent

when going from ERA5 to HIST, with average trends dropping to almost a half, and a possible reason being the different trends in the atmospheric drivers, cf. Fig. 10.

This is an epistemic problem as old as statistical climate research itself, and relates back to the concept of *perfect prognosis* (Klein et al., 1959) or in newer form to the *concept drift* in machine learning (Widmer and Kubat, 1996): every empirical

scheme has to meet the (for dynamical models almost trivial) condition that its assumptions remain valid in the predicted
future. In our case, it requires to incorporate all essential predictors and predictor-predictand relations to be stable across
different climates. There is no generally provable argument in support of the approach (observational records of the
relevant variables that could be used for verification are not long enough), and one must resort to heuristic reasoning.
With respect to applying simulated predictor fields (classifiers) it is usually assumed that their simulation is sufficiently
reliable. And as recent analyses have shown (Kendon et al., 2021), one should indeed not be too confident in our
convective classifiers *(cape, cp, tcw)* as compared to, e. g., pressure or temperature fields. Convective parameterizations, for
example, strongly depend on the native grid size, which differs markedly between ERA5 and the RCM (~31 km vs. 12 km).
This mainly affects the *cp* predictor, although no obvious trend discrepancies for *cp* are observed in Fig. 10. With respect
to a different climate, the argument is that the difference can still be seen as an anomaly from a base state and not as a
shift to a wholly new climate regime, and at least for now there is little evidence for that latter case. – Given this
uncertainty, the trend projections of this study, which were derived from a single climate model, are remarkably stable,
indicating that progress in this direction mainly lies in the dynamical modeling of convection.

**Table 3. Summary table of ETS, trends and correlations for all methods. Significant trends are boldface. For the DL methods, the ETS ensemble mean is shown.**

| model | ETS (mean) | model (ERA5) ↔ OBS annual correlation | centennial increase | | |
|---|---|---|---|---|---|
| | | | ERA5 | HIST | RCP85 2006−2100 |
| | | | 1979−2005 | | |
| LASSO | 0.46 | 0.54 | **0.25** | 0.21 | **0.07** |
| TREE | 0.52 | 0.65 | **0.34** | 0.16 | 0.03 |
| NNET | 0.47 | 0.57 | **0.32** | 0.20 | **0.07** |
| NLS | 0.44 | 0.62 | **0.31** | 0.21 | **0.05** |
| LeNet-5 | 0.46 | 0.58 | **0.27** | 0.22 | **0.07** |
| AlexNet | 0.47 | 0.59 | **0.33** | 0.18 | **0.05** |
| CIFAR-10 | 0.45 | 0.39 | **0.24** | 0.20 | **0.07** |
| ALL-CNN | 0.48 | 0.65 | **0.35** | 0.15 | **0.07** |
| GoogLeNet | 0.47 | 0.50 | **0.30** | 0.20 | **0.05** |
| ResNet | 0.48 | 0.58 | **0.30** | 0.20 | 0.04 |
| DenseNet | 0.46 | 0.51 | **0.33** | 0.20 | **0.05** |
| Simple | 0.45 | 0.46 | **0.24** | 0.22 | **0.08** |
| Logreg | 0.43 | 0.54 | **0.18** | 0.21 | **0.11** |

## 4 Conclusions

We have classified ERA5 fields of atmospheric convectivity, using an array of conventional ('shallow') and deep learning methods, with respect to the occurrence of heavy rainfall events over Germany as represented by the recently published CatRaRE catalogue. The methods ranged from very basic logistic functions to shallow neural nets, random forests (TREE) and other machine learning techniques, including the most complex deep learning (DL) architectures that were available to us. Because of the rapid progress in DL, it means we are still several years behind the state-of-the-art. Overall, we found no substantial benefit of the DL techniques over conventional (shallow) ones. On the contrary, the conventional random forest scheme TREE performed best with an ETS classification score near 0.52 for the independent validation period 2011–2020, followed by the DL networks ALL-CNN and ResNet. Those schemes seem to be best adapted for the CatRaRE classification problem presented in this study: TREE uses a clever bootstrap aggregating (*bagging*) algorithm over simple decision trees (200 in our case) whose generalization capacity is obviously crucial; and ALL-CNN and ResNet are networks of fairly moderate width and depth, for which training and testing performance are in balance. The classifiers were then applied to corresponding CORDEX simulations of present and future atmospheric fields. The resulting probabilities of convective atmospheric fields and related CatRaRE-type extreme events were increasing during the ERA5 period and also for the historic and future CORDEX simulations, independent of method. This is to be expected and in line with common wisdom of current climate research (cf. Figure SPM.6, Masson-Delmotte et al., 2021). Specifically, using TREE for the historic period (1979–2005), the resulting probabilities, measured as centennial trend, increase by 0.34 for ERA5 and by 0.16 for HIST. The discrepancy (which is less severe for the other methods) points to modeling inadequacies with respect to convective environments, as suggested by Fig. 10. For the future CORDEX simulations we obtained a significant increase of around 0.07 for most methods, the number being smaller likely because of applying a linear trend to (bounded) probabilities. The overall tendency towards more extreme convective sub-daily events is consistent with recent estimates from Clausius-Clapeyron temperature scaling (Fowler et al., 2021) as well as from a convection-permitting dynamical climate model for Germany (Purr et al., 2021).

Compared to other classification problems such as the notorious image classification contest ImageNet, our setup of a binary classification is quite simple. One must keep in mind, however, that the very design of CNNs, with their focus on 'features' of colored shapes (objects), is modeled along the lines of ImageNet and relatives. Applying a CNN to other, not object-like 'images' (blurred boundaries and colors) is not guaranteed to work out of the box. But it does, as we have seen, with only moderate adjustments. The main difficulty here was to understand just how much quicker the more complex models would learn, so that we had to shorten their learning period considerably to avoid overfitting.

One may object that by choosing all of Germany as the (uniform) study area our approach misses important regional detail, leaving only little relevance for local decision makers. The study is nevertheless the first of its kind to actually estimate future statistics of CatRaRE-type events from convective atmospheres, and should contribute to raise awareness among researchers and decision makers for an impending change in these statistics. Given the wealth of methods, regional detail would at this point just add another strain to deal with, so we decided against it. Yet, it remains a valid subject for

prospective studies. This is to be done with further refinements, the ultimate goal being the classification and projection of impact-relevant convective rainfall events for as small a region as the setting allows. So far the only criterion to isolate convective events from the CatRaRE database was their duration (here 9 hours). By considering more than two classes, e. g. by introducing more regional and temporal detail, or more levels of intensity, the full power of CNNs, and here perhaps of ALL-CNN or ResNet, could be exploited. That way, the usefulness of the results for decision makers in risk management could be increased substantially. It is hoped that adding truly multivariate, pattern-based atmospheric predictors, such as moisture convergence or vorticity, can foster the performance especially of CNNs with their feature extracting capabilities. And there is a good chance that with all these refinements especially the DL methods, which are designed to handle considerably more complex classification targets, remain sufficiently reliable.

But for now our conclusions entail in passing that from this study, deep learning methods are not surpassing the conventional ('shallow') statistical toolbox. It will be interesting to follow the evolution in state-of-the-art dynamical models. Specifically, how does the development of convection-permitting dynamical models (e. g. Kendon et al., 2021) compare to DL-based convection schemes (e. g. Pan et al., 2019)? And why should their integration not offer the best of both worlds in one (Wang and Yu, 2022; Willard et al., 2022)?

## 5 Code availability

The relevant code underlying this paper can be found at https://gitlab.dkrz.de/b324017/carlofff and is archived at Zenodo (https://zenodo.org/record/8146270). Training and deployment of DL models is performed using the *Caffe* framework with its Octave interface (https://github.com/BVLC/caffe).

## 6 Author contribution

GB and MH designed the experiments and GB carried them out, developed the model code, performed the simulations and prepared the manuscript with contributions from MH.

## 7 Competing interests

The authors declare that they have no conflict of interest.

## 8 Acknowledgements

We enjoyed fruitful discussions with Georgy Ayzel. The study was funded via the "ClimXtreme" (sub-project CARLOFFF, grant number 01LP1903B) by the German Ministry of Education and Research (Bundesministerium für Bildung und

Forschung, BMBF) in its strategy "Research for Sustainability" (FONA). This work used resources of the Deutsches Klimarechenzentrum (DKRZ) granted by its Scientific Steering Committee (WLA) under project ID bb1152.

## 9 Declaration

The authors declare that they have no conflict of interest.

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

495

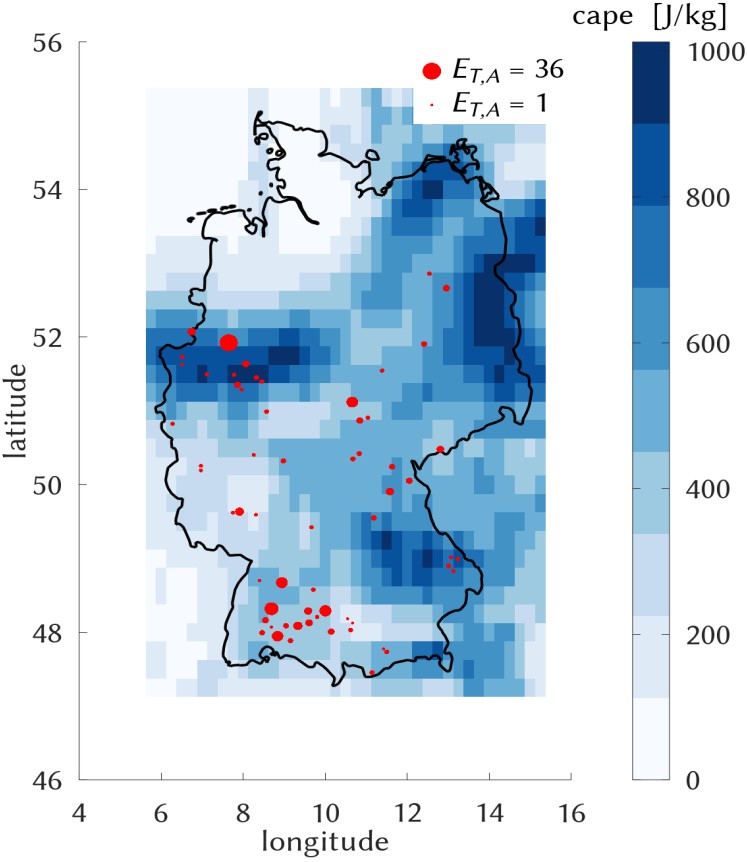

**Figure 1. The conditions for cape on July 28, 2014 (blue), along with $E_{T,A}$ values of corresponding CatRaRE events of ≤9h duration (dots).**

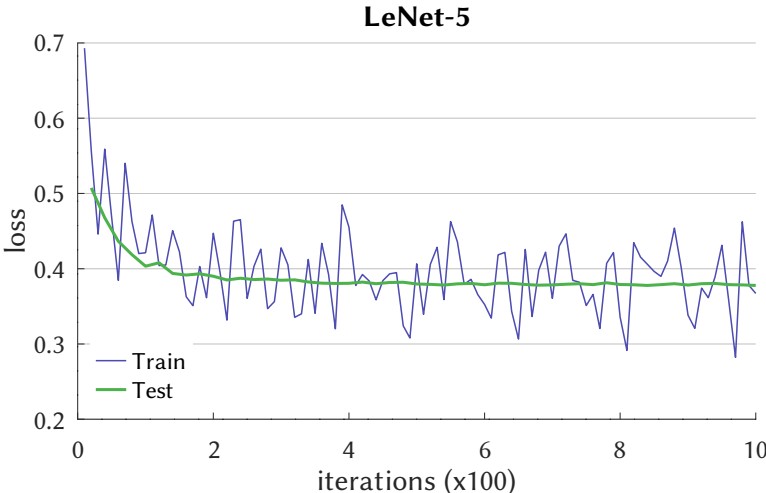

**Figure 2. Learning curve of the LeNet-5 network, with crossentropy as loss. Iterations indicate the number of batch passes (batch size $k$=100). Testing uses averages of roughly $n/k$ batches, $n$ being the calibration time series length.**

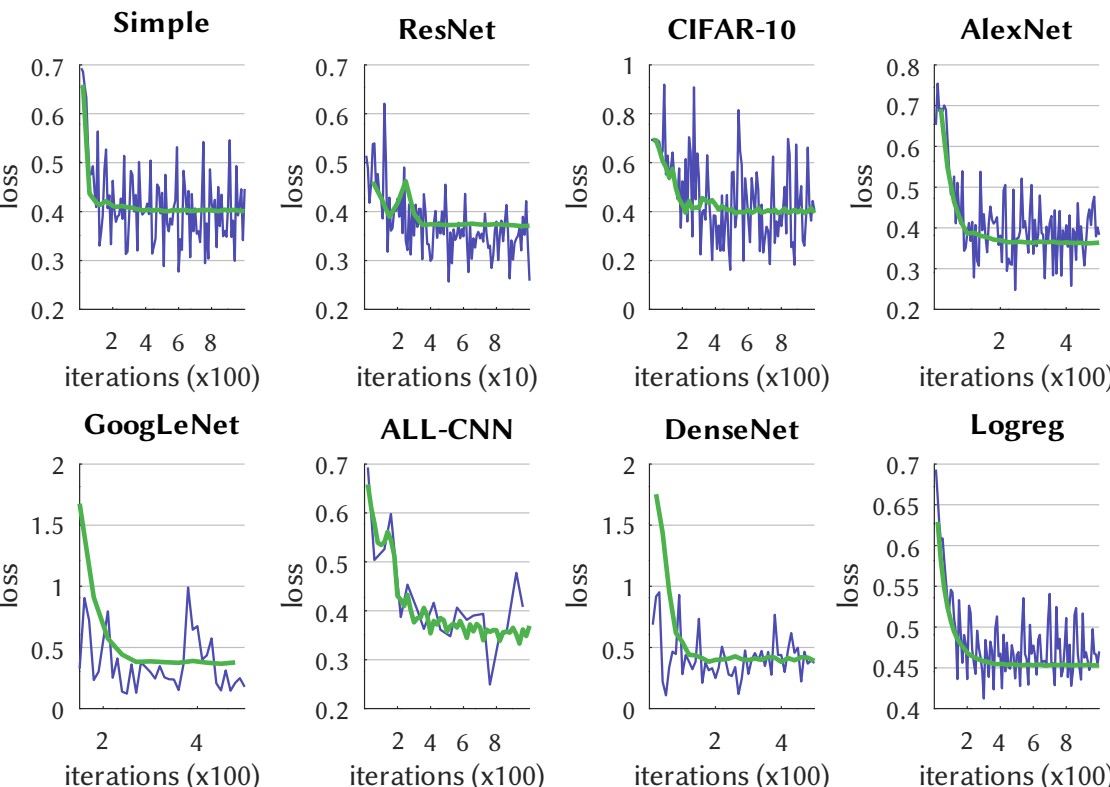

**Figure 3. As Figure 2, for the other DL networks (using blue for Train and green for Test).**

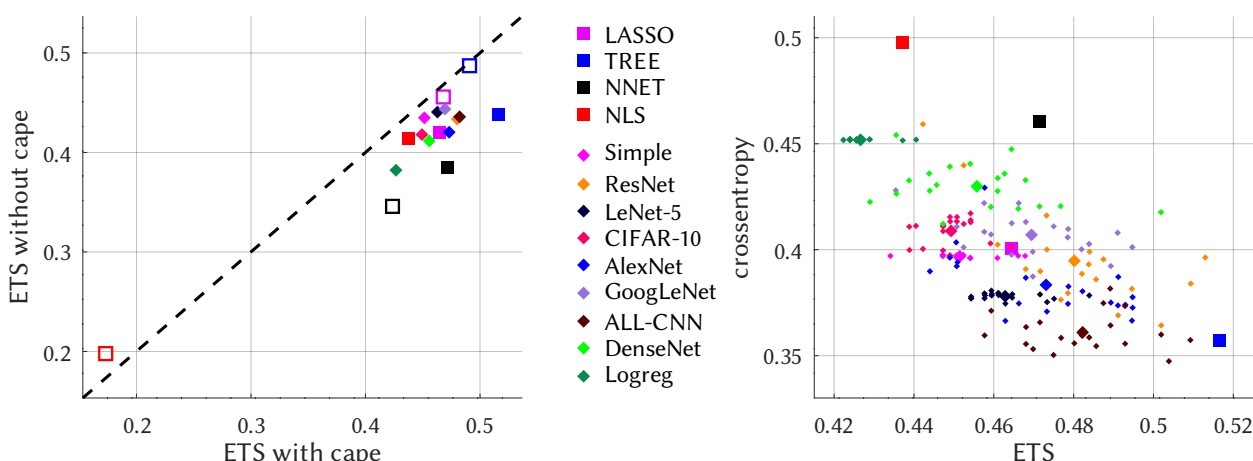

**Figure 4. Model performance for the validation period 2011–2020. Left: ETS with and without cape as a predictor. Right: Relation between ETS and crossentropy (both with cape). Squares depict Shallow, diamonds Deep models. Unfilled markers in the left panel symbolize no EOF truncation.**

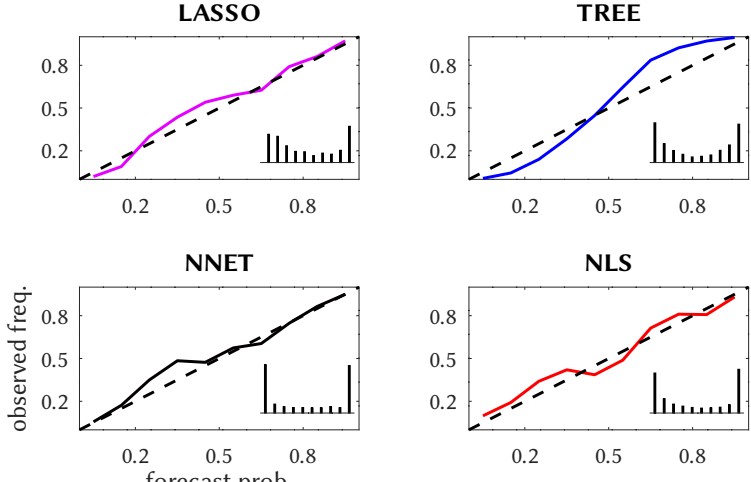

**Figure 5. Reliability diagram for the shallow methods, with forecast histogram inset based on 10 bins and constant y-axis scale.**

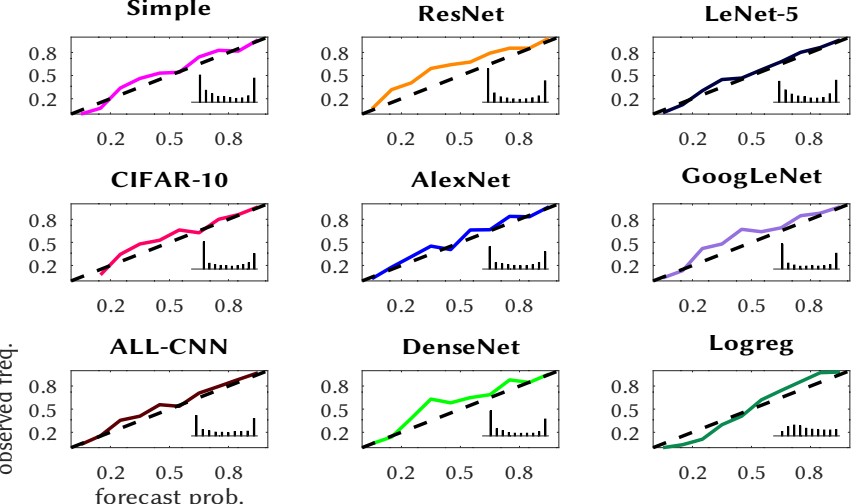

**Figure 6. Like Fig. 5, for the deep methods.**

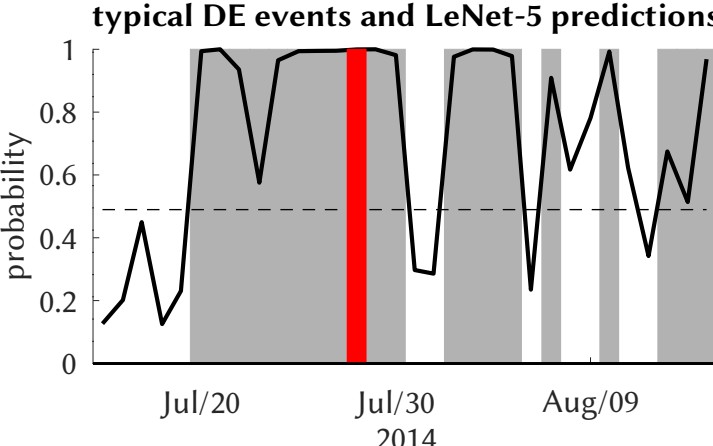

**Figure 7. Typical probability output of the LeNet-5 model (black) with threshold (dashed, see text) around the July 2014 event (red); other events are gray.**

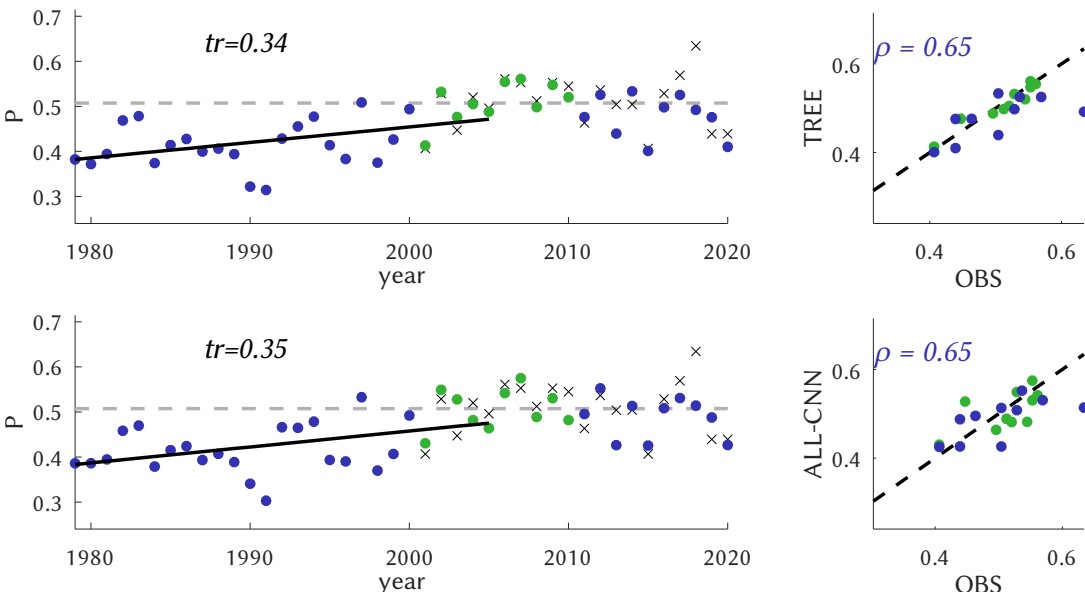

**Figure 8. Annual values of the probability P of CatRaRE-type events, as observed (crosses) or simulated from ERA5 (dots), using TREE (top) and ALL-CNN (bottom); the calibration period is marked as green and the rest as blue. The 1979–2005 time period reveals a significantly positive trend for both models, displayed as ΔP/100y; observed 2001–2020 climatology (gray dashed) is given for reference. The scatterplots on the right-hand side depict the same data as a scatterplot against observations, with correlations for the validation period.**

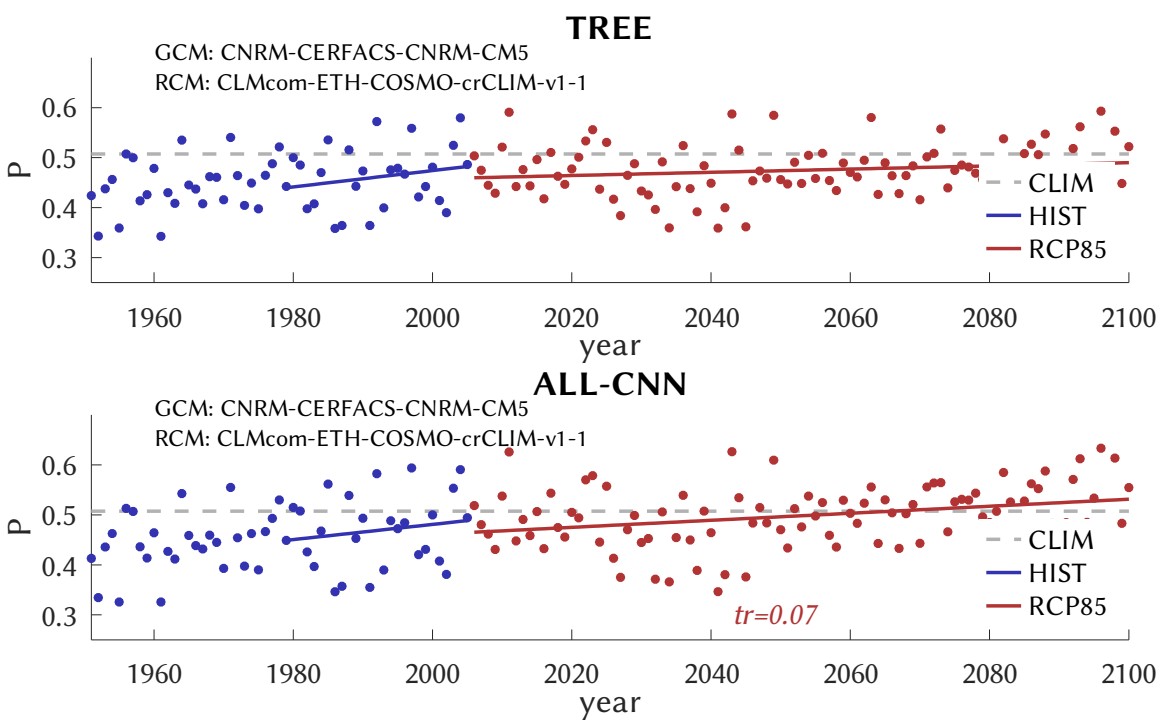

**Figure 9. Similar to Figure 8, as simulated by GCM/RCM, for historic (blue) and future (red) emissions. For reference, the observed 2001–2020 climatology is also shown (CLIM, gray dashed).**

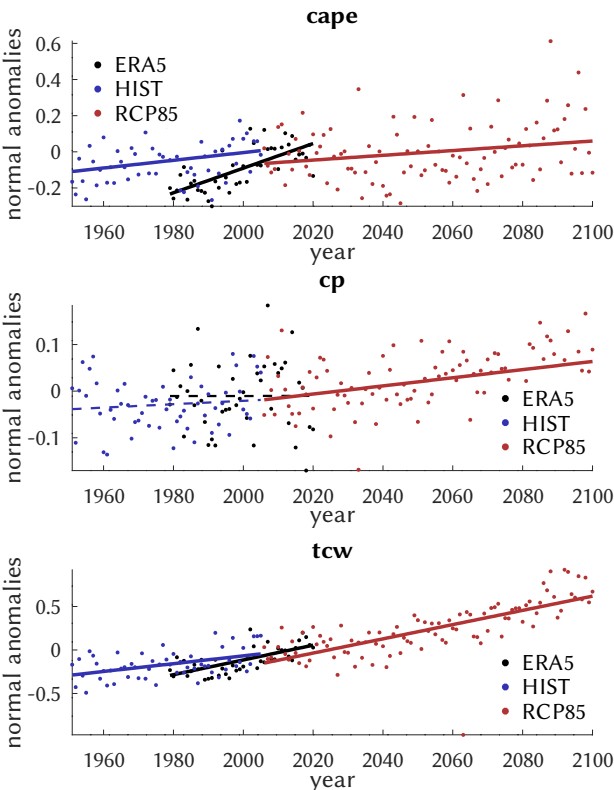

**Figure 10. Similar to Fig. 9, for the predictor series and including ERA5 (black) as they enter the classification (normalized). The trends pertain to the full available series, and significance is indicated by solid vs. dashed lines.**