# Peer review of "Shallow and deep learning of extreme rainfall events from convective atmospheres"

_EGUsphere, 2022_

## Author Response (AR1)

Here we substantiate our first response given to reviewers A and B at https://egusphere.coperni-cus.org/preprints/2022/egusphere-2022-1159. Line numbers refer to the new text.

We note that due to an error in the ETS estimate (the probability threshold for validation was optimized and not taken from the calibration set), the corresponding results have shifted so that now TREE has the best ETS score.

**Rev #1**

The manuscript is well structured, and I appreciate the extensive model selection used for comparison and acknowledge the effort spent to train all of these. Even though the intercomparison of different methods and architectures is interesting on its own, I have difficulties distilling the overall relevance (concrete use case) of the classification for meteorological applications.

Please see the new § (l. 62-73) which explains more clearly our intention to "to raise awareness among researchers and decision makers for an impending change in these statistics".

**Major Comments**

- As mentioned above, it does not become clear to me what consequences a statement like "There is an extreme convective event (somewhere) over Germany" might have for a meteorologist, climatologist or decision-maker. L 229f somehow reflects the ultimate goal; however, it might be good to further distil the gain also in the introduction.

See comment above and also l. 279-287.

- I wonder how a cross-entropy or ETS analysis might contribute to a better understanding of the influence of 'deep' in DL models, as stated in l. 41f. For such a statement, I would have expected some explainable AI (XAI) methods or some sensitivity analysis of each model type, like varying the number of inception blocks in the 'GoogLeNet-style' model. Here the introduction raises expectations that the conclusion does not reflect.

We give our interpretation of the results depending on network architecture in the new § at l. 194-212. This includes a discussion on width and depth of a network and if extended complexity is necessary in our case or not.

- As far as I understand, you are using ERA5 data (cape, cp, tcw) as input **X** and CatRaRE as target **y** for training (2001-2010) and validation (2011-2020). Finally, you apply the trained model to data from HIST and RCP85. In l 144, you correctly state that the second dataset is not independent of the DL models, as you use those for model selection. As overfitting can happen on both - parameters (training set) and hyperparameters (validation set), why do you not split your data into three sets (training, validation, test)? Especially as you apply the trained models to data from different sources that likely have different properties, I think it would be beneficial to compare the test set's performance against the same (sub-)period of RCP85. Thus, you could detect differences in model performance that

might serve as a guide towards interpreting all RCP85 data where you do not have any labels.

This topic was already discussed in the discussion phase, where we argue that overfitting is not to be expected; see also §2.4 of the text.

- I suggest broadening the analysis of the predicted probabilities over the entire detection period. For example, replacing Fig. 5 with a reliability diagram where the predicted probability is plotted against the observed relative frequency might reveal model-specific differences.

We have added reliability and sharpness in Figs. 5 and 6, plus the new §3.3.

- Given the close range of ETS values across the different models, I suggest providing uncertainty quantifications and/or statistical tests to demonstrate the significance of your findings.

Stochastic uncertainty in shallow modeling was missing in the discussion paper. We have now reformulated TREE and NNET as ensemble models and consider, like in the DL case, corresponding means. The SI presents a second ('cloned') realization of all models demonstrating that the results are essentially stable with respect to the main conclusions, l. 189-193.

- How do already existing 'classical' findings of the expected change of extreme precipitation align with your classification results? Can you discuss the concept drift in the data that the classifier faces?

We have expanded on the "common wisdom" as reported by the cited IPCC source, see l. 291-301. For the concept drift we have a new § at l. 266-278.

- In that regard, which period do you use to calculate the mean and std for the z-transformation?

It is 2001−2020, as mentioned in l. 91.

**Minor Comments**

- L. 22ff Besides the references to the 'classical' DL introductions, I encourage the authors to also focus on the recent discussions on ML/DL applications in atmospheric sciences like Reichstein et al. (2019) and Schultz et al. (2021).

References are added.

- L. 158f How do you analyse the influence of cape? In l. 126 you state that you are using cape, cp and tcw as channels similar to RGB. Please clarify how you create the "non-cape" classifications. Do you train the models with two channels only? Do you replace the cape channel with zeros or another variable?

We added a corresponding footnote at l. 178.

- Fig. 1 shows cape values jointly with the CatRaRe events used to define the extreme labels. The selected model domain contains pixels outside of Germany. CatRaRE, however, covers Germany only. Did you check (most likely with some other dataset) how often (if at all) extreme events occur outside of Germany but within your defined model domain? For me, that seems to be a potential source of introducing labelling errors.

Here we disagree, the German border has no relevance for the classification. If a pixel falls outside, it may only mean that it is too distant to affect the local event, a fact that should be learned by the schemes.

- Fig. 4: I suggest using a more colourblind-friendly palette.

We have put the updated Fig. 4 through Coblis and it looks good there.

- Even though Table S1 lists several tuned hyperparameters, how does the learning rate change under the poly policy?

The hyperparameters are described in the SI. The learning rate follows a polynomial decay, $(1 - iter/max\_iter)^{power}$, becoming zero when max_iter is reached. This is added as a footnote on l. 137.

- I suggest adding a column reporting the number of trainable parameters of your modified versions

We have added a corresponding column to Table 2 (which describes the network architecture).

- Did you consider also using architectures already focussing on precipitation (for example (your) RainNet model (Ayzel et al., 2020)) and adjusting details for your classification task?

No. RainNet is used to map one state of a system to another of the same system, whereas here we need to map one state of one system to another of another system. Or, in other words, RainNet was designed to capture the motion and intensity dynamics of precipitation fields at very high spatial and temporal resolution, and required much larger amounts of data for training (several years of five minute data). We would not consider RainNet as specifically suitable just because it aims at the same variable, precipitation. The processes or relationships learned by RainNet are very different from our setup. We do not say that RainNet is unsuited for the task, but our approach of selecting candidate DL models was a different one (using established models for image classification).

- L. 59 I am wondering if a log transformation for cp before applying the standardisation might be beneficial

We discuss this in a footnote on l. 87.

- Please provide some more details on the EOF reduction. For example, how many components are you using?

This was added at l. 117.

- From the first sentence in your abstract, I expect this manuscript to focus on creating a new data set that can be used for ML/DL applications. In its current state, the abstract does not adequately transport the enormous (DL-)model comparison you performed.

We changed the abstract accordingly.

**Formal Comments**

- Please add a "competing interests" statement as required by Copernicus Publication (see https://www.natural-hazards-and-earth-system-sciences.net/submission.html#manuscript-composition §16)
- Software Code: You refer to your GitHub repository but to the best of my knowledge Copernicus Journals prefer software provided through a DOI (e.g. through zenodo)
- URLs: Please add the last access dates to all URLs
- A legend is missing in Fig. 3

All have been addressed, except: Our repository is gitlab, to which unfortunately no connector from Zenodo exists; there are no urls in the References; legend info added in the caption.

**Rev #2**

We would like to thank the referee for taking the time to review our manuscript, and for the open and clear criticism. The central part of the comment is, as we understand, as follows:

*„[...] [the manuscript] basically [...] is not driven by any research questions and objectives. Without finely tuning each method, especially for the sophisticated CNN models, it is unclear why to compare these methods and also the very similar results for each method give readers very limited insights from their studies."*

We have revised the manuscript to better convey the research questions of our study. Please see the tracked changes, in particular sections 1, 3.2, 3.4 and 4.

The results pertaining to the DL models are inconclusive because their application requires more fine-tuning.

We repeat that with regard to DL tuning, we have described that we purposefully used the basic model structure *as is* from the corresponding image recognition tasks, but fine-tuned settings to achieve convergence in the learning curves. Model uncertainty, moreover, is a genuine part of DL and thoroughly covered by Brownlee (2018), we have addressed it in greater detail in the abstract, §3.2 and §4.

The results are not relevant to the reader because the performance of all benchmarked models is very similar.

While we do not understand the argument (as addressed in the paper discussion), after fixing the above-mentioned error the results are no longer very similar (and have not been before).

---

## Referee Report (RR1)

The paper by Bürger and Heistermann applies a variety of machine learning algorithms to the task of classifying atmospheric fields as conducive to severe convection in Germany. Trained and evaluated on ERA5 and CatRaRE, the models are applied to EURO-CORDEX simulations for the past and future to analyze trends in the potential for severe convection.

This paper stands out from the mass of applied machine learning papers by actually comparing a full range of simple, intermediate and very complex models. Given the great recent interest in such methods, I think that this kind of study can be valuable to the natural hazards community, as well as climate science in general. The paper is overall sufficiently well written and the methodology seems sound to me. I nevertheless have a number of concerns with the current state of the manuscript, which require some further revision before I recommend its publication.

**General comments:**

- The motivation for this work should be clarified in the introduction: Make it clear that it is far from obvious that the "killer apps" from image classification contests will be similarly worth their cost in a weather and climate context. I see a number of reasons for doubting their applicability:

    - atmospheric fields have very different spatial statistics from images of cats and dogs

    - your target quantity (severe convection or not) exists on a spectrum. As far as I'm aware, there are no animals that are "close to the threshold" of being a cat or dog.

    - the amount of training data is often severely limited in our field

    - unlike MNIST and the like, your "images" might have long term trends (see also my comment below)

    It is therefore a good idea to actually test whether there is any benefit in the "deeper" approaches over classical machine learning / statistics.

- The conclusions should be made more clear as well. I would argue that you found no substantial benefit of deep over shallow methods (NNET for example seems competitive with its deeper sibling models but is so much simpler). That is an interesting result and arguably good news for researchers with limited expertise in deep learning who can thus rely on relatively cheap and simple methods.

- To what extent do you think that your conclusions can be generalized to other similar studies? I guess one thing to keep in mind is that hazards are usually rare whereas you define your problem in a way that leads to a balanced dataset.

- I have several comments on the issue of trends in your data:

    - It is well known that neural networks are typically bad at extrapolating

outside of the training range. This could play a role here, depending on how exactly you normalized the data from ERA5 and the CORDEX simulations. Did you estimate the mean and standard deviation from the whole period? Or just the training and / or validation time?

- ○ Are the trends consistent across your 20 optimization runs or do they randomly differ?

- ○ Did you look at the trends in your predictor variables? If ERA5 and CORDEX have different trends in, for example, CAPE, couldn't that easily explain the discrepancy between HIST and ERA5 trends?

- ○ For trends in ERA5 convection parameters, you can also refer to https://doi.org/10.1038/s41612-021-00190-x

- I think that "explainable AI" methods like Shapley values (which are model agnostic, see https://doi.org/10.48550/arXiv.1705.07874 ) would have been helpful in actually understanding the differences between the different models. This could have explained which model uses which variables and which of the variables are responsible for the trends. It is likely too late to add such analyses to the manuscript, but it could be mentioned in the conclusion / outlook section and perhaps taken into account in future studies.

- I am a little concerned about the use of convective precipitation (cp) in this study. The amount and spatial distribution of rainfall produced by the convection parametrization depends heavily on (a) the type and settings of the parametrization and (b) the internal horizontal resolution of the model. The resolution definitely differs between ERA5 and COSMO. I'm not sure how similar the parametrizations are, I guess they are both based on the Tiedtke scheme? In light of possible fundamental differences, how can you apply a model trained on ERA5 to cp fields from another model? Wouldn't it be wiser to use total precipitation instead, which is more generally comparable?

- Is there a reason why no dynamical variables like wind shear were included?

**Specific comments:**

l.18 and other places: please explain somewhere, what exactly you mean by "saturation effect".

l.62 "the best performing models are applied" I believe you apply all of them, which is a good idea.

l.92 "RCP85 (2006-2020)" didn't you use the full scenario run from 2006-2099? Or does this sentence only apply to the climatology used as reference? Either way please clarify over which part of the time series you estimated the mean and standard deviation for your normalization in each case. In particular when applying your models to the full ERA5 time series, did you use the same normalization as in training or normalize over that whole period? I think this

could make a difference for the resulting trends.

l.114: "we employ four shallow statistical models:" but then you list only three, forgetting NLS

l.116 please explain what exactly you mean by EOF orthogonalization. Did you apply eofs to the fields and use the principal components as predictors? If so, which part of which time series were the EOFs estimated from?

l.116 How did you arrive at the seemingly random number of "33, 27 and 21" EOFs?

l.124 How is "Logreg" a deep method with only one layer? And how does it differ from your logistic regression (NLS)?

l.126 please upload the code to some permanent repository like zenodo, as per journal policy. I don't know what you mean by "no connector from Zenodo exists", why can't you just upload your code there?

l.152 what do you mean by "EOF truncation"? The same as EOF orthogonalization above?

Fig.2 and 3: why is the test loss so much smoother than the training loss?

l.175 what do you mean by "optimal probability threshold"? Don't you just predict a class if its probability is greater than 0.5?

Fig.4 please explain why you are specifically interested in the importance of cape? what conclusions do you draw from this?

l.268 "reasons unknown" is it so surprising that convective activity would have different trends in ERA5 and COSMO?

Table 3: Is it correct that none of the HIST trends are significant (not boldface) despite their relatively large magnitudes?

---

## Author Response (AR2)

We thank both reviewers for their time and effort to provide their very helpful comments. Here we address the major ones (the minor ones having entered the revision without extra notice).

**Reviewer #1**

I want to thank the authors for their time and effort in revising their manuscript. While the authors adequately addressed most of my previous remarks, there are still some open issues, so I'd like to ask for more clarification. I still have some mixed feelings regarding a recommendation for publishing. Those mixed feelings are primarily due to the "unclear" study motivation (see below), making it hard to distil scientific significance. Jointly with the gap to state-of-the-art models, which become easily accessible through, for example, Hugging Face (https://huggingface.co/) I would recommend a major revision of the study's motivation.
Please note that I don't see the gap to the latest methods as an exclusion criterion for publication, but the motivation and, thus, the significance of the study must be better and more clearly explained.

Study's motivation:
I have some difficulties following your revised motivational paragraphs. I know that answers from the discussion phase might develop further until the revised version is submitted, especially when larger parts are modified or extended. However, in its current state, the focus lies more on what the study is not, instead of -for example - the occurrence-frequency modelled by the different approaches, as you mentioned as an answer in the discussion phase. Moreover, I think that statements about further studies (l. 69) are better suited in the discussion/outlook section rather than as a motivation for your current study.

We have thoroughly reworked the introduction. In short, we put the research now in the general framework of sub-daily downscaling of extreme, impact-relevant precipitation events. Convective indices require high-resolution atmospheric fields, which considerably limits future scenarios. Corresponding impact studies have to our knowledge not been conducted so far.

L70f Even though you present your interpretation of model architecture etc. (lines 194-212), I find the phrasing of the influence misleading, as I would still expect some XAI /sensitivity analysis later on.

See comment above: the entire context has been rewritten.

L190ff: Did you consider using a SHAPE-value (Lundberg and Lee, 2017) analysis for your (shallow) models?

Since we only use 3 convective indices as predictors, and because we did not want to overload the analysis with too much detail, we decided against using an extra SHAP analysis. For the shallow models, moreover, the relevant configuration uses an EOF filter, so that SHAP would not reveal the direct influence of the indices anyway. Note that the influence of *cape* had been established already (cf. Fig. 4), and this revision additionally provides the centennial trends of all indices in Fig. 10.

L176f What are those optimal thresholds? How did you determine them, and do they vary between models?

The optimal probability thresholds are determined by maximizing the calibration skill; they vary between 0.42 (Lasso) and 0.59 (NLS). A clarification is added (l. 215).

L176f Did you also consider using the Brier Score, which directly accesses forecasted probabilities?

No, but the Brier score is closely related to crossentropy, both measuring the deviance between predicted and observed (post-fact) probabilities.

Fig. 1.: I can not fully follow your line of argumentation, so I am trying to clarify: I fully agree that the German border should not have any influence on the classification itself (especially the CNN-based classification models are in search of whether a specific pattern (extreme event) is present or not. However, my point of concern was related to the initial labelling. You are using CatRaRe events within Germany for labelling the selected ERA5 domain that also covers parts outside of Germany. So as far as I understand the labelling procedure, an extreme event occurring in - say, in the western parts of the Czech Republic would not be present in CatRaRe, and thus it would be labelled as "not extreme". As you are aiming to solve a classification task and not an object detection task (creating something like a bounding box, see, for example, YOLO (https://arxiv.org/pdf/1506.02640v5.pdf), the same or at least very similar ERA5 patterns (independent of their geographic position) might be labelled differently, which in turn, might negatively affect the classification in the next step. Have you looked closely into events where your models predicted an extreme that wasn't observed?

The approach must be understood in terms of transfer functions and downscaling, the main (sole) criterion being explained variance or similar *skill* for a given set of local observations, a typical example being (Harpham & Wilby, 2005, cf. Fig. 1). And since rainfall may happen to occur slightly off any one of the stations, your argument equally applies to that case and in fact to any station downscaling. We have added a corresponding cautionary note (l. 131).

L242 I need clarification on which additional information is conveyed by Fig. 7, especially as the mentioned (see above) optimal probability thresholds are not shown. Is the intention showing the predicted high probability on the 29th of July? But what happens if I select the 25th(?) of July (forecasted probability ~0.5) and the 7th(?) of August (forecasted probability ~0.6)? Given the identical probability threshold, one event would be misclassified in that example. Did you intend to somehow cross-check for detecting the "most famous" convective extreme event(s)? If so, please clarify. Otherwise, one might get the impression of cherry-picking.

We have added the appropriate threshold, as determined during calibration. Note, however, that our original intention was to just convey a first intuition about the probabilistic forecasts, which are independent of the threshold.

**Reviewer #2**

The paper by Bürger and Heistermann applies a variety of machine learning algorithms to the task of classifying atmospheric fields as conducive to severe convection in Germany. Trained and evaluated on ERA5 and CatRaRE, the models are applied to EURO-CORDEX simulations for the past and future to analyze trends in the potential for severe convection. This paper stands out from the mass of applied machine learning papers by actually comparing a full range of simple, intermediate and very complex models. Given the great recent interest in such methods, I think that this kind of study can be valuable to the natural hazards community, as well as climate science in general. The paper is overall sufficiently well written and the methodology seems sound to me. I nevertheless have a

number of concerns with the current state of the manuscript,which require some further revision before I recommend its publication. General comments:

● The motivation for this work should be clarified in the introduction: Make itclear that it is far from obvious that the "killer apps" from image classification contests will be similarly worth their cost in a weather and climate context. I see a number of reasons for doubting their applicability:

○ atmospheric fields have very different spatial statistics from images of cats and dogs

○ your target quantity (severe convection or not) exists on a spectrum. As far as I'm aware, there are no animals that are "close to the threshold" of being a cat or dog.

○ the amount of training data is often severely limited in our field

○ unlike MNIST and the like, your "images" might have long term trends (see also my comment below) It is therefore a good idea to actually test whether there is any benefit in the "deeper" approaches over classical machine learning/ statistics.

We have thoroughly reworked the introduction touching all points above (see response to rev. #1). See especially the new § at ll. 71–87.

● The conclusions should be made more clear as well. I would argue that you found no substantial benefit of deep over shallow methods (NNET for example seems competitive with its deeper sibling models but is so much simpler). That is an interesting result and arguably good news for researchers with limited expertise in deep learning who can thus rely on relatively cheap and simple methods.

We have added a corresponding sentence in the conclusions (l. 354). We see the NNET performance slightly different, however. From Fig. 4 it seems clear that at least ALL-CNN and ResNet outperform the simple NNET, especially given the fact that hyperparameter tuning was only moderate.

● To what extent do you think that your conclusions can be generalized to other similar studies? I guess one thing to keep in mind is that hazards are usually rare whereas you define your problem in a way that leads to a balanced dataset.

Obviously, there is no catch-all method to deal with highly imbalanced data. Going from a gridpoint-based approach to areal averages trivially increases scores at the cost of regional detail. For the former, for example, Ukkonen and Mäkelä (2019) get fairly weak scores around 0.2 (area under precision-recall) which may partly put their gridpoint results into question. While we obviously cannot give advice for the general case, it will be one of the main tasks for a subsequent study to find a good balance for the case of CatRaRE, and how this translates into projection uncertainty.

● I have several comments on the issue of trends in your data:

○ It is well known that neural networks are typically bad at extrapolating outside of the training range. This could play a role here, depending on how exactly you normalized the data from ERA5 and the CORDEX simulations. Did you estimate the mean and standard deviation from the whole period? Or just the training and / or validation time?

Normalization is indeed important since many of the tools, especially the neural networks, are quite sensitive to biases. For downscaling, the adopted standard is to normalize the respective fields of reanalyses on the one hand and simulations on the other to their own base climate. As described in §2.1 the common period 2001–2020 is used.

○ Are the trends consistent across your 20 optimization runs or do they randomly differ?

Trends were unfortunately not saved during the training phase. For the application part, to save computation time and avoid an overload of results, only the best performing members were used for getting the trends (see l. 232). To answer the question nevertheless, we have done a few extra simulations. They show small variations in the main trends that are to be expected from the training uncertainty, but the main message regarding the significant trends remains valid. We have added a corresponding clarification.

○ Did you look at the trends in your predictor variables? If ERA5 and CORDEX have different trends in, for example, CAPE, couldn't that easily explain the discrepancy between HIST and ERA5 trends?

We have included centennial trends for all indices now in Fig. 10, and it possibly answers your question regarding the ERA5/HIST discrepancy of trends. Thanks!

○ For trends in ERA5 convection parameters, you can also refer to https://doi.org/10.1038/s41612-021-00190-x

We added that, along with Lepore et al. 2021.

● I think that "explainable AI" methods like Shapley values (which are model agnostic, see https://doi.org/10.48550/arXiv.1705.07874 ) would have been helpful in actually understanding the differences between the different models. This could have explained which model uses which variables and which of the variables are responsible for the trends. It is likely too late to add such analyses to the manuscript, but it could be mentioned in the conclusion / outlook section and perhaps taken into account in future studies.

See our comment to rev. #1 regarding SHAP values.

● I am a little concerned about the use of convective precipitation (cp) in this study. The amount and spatial distribution of rainfall produced by the convection parametrization depends heavily on (a) the type and settings of the parametrization and (b) the internal horizontal resolution of the model. The resolution definitely differs between ERA5 and COSMO. I'm not sure how similar the parametrizations are, I guess they are both based on the Tiedtke scheme? In light of possible fundamental differences, how can you apply a model trained on ERA5 to cp fields from another model? Wouldn't it be wiser to use total precipitation instead, which is more generally comparable?

This is a valid criticism, and we have added an extra § in the discussions (l. 323–329, 337–339). Using cp was guided by our overall subject of convection.

● Is there a reason why no dynamical variables like wind shear were included?

No, there is no particular reason other than just using the indices that were directly available from the ESGF, and wind shear was not. As e. g. compared to Ukkonen and Mäkelä (2019) who use a total of 40 predictor variables (including wind shear), our focus lay on a rather simple predictor setting, knowing that this would entail, among other things, a greater need for explainable AI or similar.

Specific comments:
l.18 and other places: please explain somewhere, what exactly you mean by "saturation effect".

replaced, see ll. 291, 303, 367.

l.62 "the best performing models are applied" I believe you apply all of them, which is a good idea.

Fixed.

l.92 "RCP85 (2006-2020)" didn't you use the full scenario run from 2006-2099? Or does this sentence only apply to the climatology used as reference? Either way please clarify over which part of the time series you estimated the mean and standard deviation for your normalization in each case. In particular when applying your models to the full ERA5 time series, did you use the same normalization as in training or normalize over that whole period? I think this could make a difference for the resulting trends.

We use the full 2006–2100 scenario. The corresponding part now reads: *"The atmospheric fields are given as normalized anomalies, using mean and standard deviation of the reanalyzed and simulated fields from the common period 2001–2020 as a general reference state; for the latter, it requires to concatenate the corresponding sections from HIST (2001–2005) and RCP85 (2006-2020) to form the reference."*

l.114: "we employ four shallow statistical models:" but then you list only three, forgetting NLS

Fixed.

l.116 please explain what exactly you mean by EOF orthogonalization. Did you apply eofs to the fields and use the principal components as predictors? If so, which part of which time series were the EOFs estimated from?

Yes. EOFs are formed from the calibration period and the fields are projected onto those, a procedure usually called 'EOF truncation' and also adopted here now. Clarified in the text (l. 148).

l.116 How did you arrive at the seemingly random number of "33, 27 and 21" EOFs?

We use North's rule of thumb. Added in text (l. 148).

l.124 How is "Logreg" a deep method with only one layer? And how does it differ from your logistic regression (NLS)?

Yes, Logreg is doubtlessly anything but deep. And it is also similar to the NLS. The difference lies in the loss function (crossentropy vs. squared errors) as well as the optimizing algorithm. NLS works deterministically on the entire calibration set, using the Levenberg-Marquardt algorithm; Logreg, on the other hand, employs the stochastic gradient decent optimizer with the Adam solver. From Fig. 4 it is apparent that, unlike Logreg, NLS (without EOF) does not find a useful optimum. 'Logreg' and 'Simple' are mainly used for checking the caffe software and its solver *Adam*. We have added some clarification (l. 157).

l.126 please upload the code to some permanent repository like zenodo, as per journal policy. I don't know what you mean by "no connector from Zenodo exists", why can't you just upload your code there?

We now have additionally archived the code at Zenodo (cf. §5).

l.152 what do you mean by "EOF truncation"? The same as EOF orthogonalization above? Fig.2 and 3: why is the test loss so much smoother than the training loss?

We use now EOF truncation throughout the paper, see above. Test loss is the result of averaging over as many (independent) batches as fit in the calibration set. Clarified in the caption of Fig. 2.

l.175 what do you mean by "optimal probability threshold"? Don't you just predict a class if its probability is greater than 0.5?

No, we use thresholds that optimize the calibration ETS (l. 215).

Fig.4 please explain why you are specifically interested in the importance of cape? what conclusions do you draw from this?

We have added a comment and source for the influential role of cape as a predictor (l. 217).

l.268 "reasons unknown" is it so surprising that convective activity would have different trends in ERA5 and COSMO?

This point is now void through the provision of the direct predictor trends.

Table 3: Is it correct that none of the HIST trends are significant (not boldface) despite their relatively large magnitudes?

Yes, the likely reason being the shortness of the period.

**References**

Harpham, C., & Wilby, R. L. (2005). Multi-site downscaling of heavy daily precipitation occurrence and amounts. *Journal of Hydrology*, *312*(1–4), 235–255. https://doi.org/10.1016/j.jhydrol.2005.02.020

Ukkonen, P., & Mäkelä, A. (2019). Evaluation of Machine Learning Classifiers for Predicting Deep Convection. *Journal of Advances in Modeling Earth Systems*, *11*(6), 1784–1802. https://doi.org/10.1029/2018MS001561